# Disability discrimination and well-being in the United Kingdom: a prospective cohort study

Ruth A. Hackett [ID] ,[1,2] Andrew Steptoe,[2] Raymond P. Lang,[3] Sarah E. Jackson [ID] [2]

[1]Health Psychology Section, King's College London, London, UK
[2]Behavioural Science and Health, University College London, London, UK
[3]Leonard Cheshire Research Centre, Epidemiology and Public Health, University College London, London, UK

**Correspondence to**
Dr Ruth A. Hackett;
ruth.hackett@kcl.ac.uk

## ABSTRACT

**Objectives** Disability discrimination is linked with poorer well-being cross-sectionally. The aim of this study was to explore prospective associations between disability discrimination and well-being.

**Design** Prospective cohort study.

**Setting** The United Kingdom Household Longitudinal Study.

**Participants** Data were from 871 individuals with a self-reported physical, cognitive or sensory disability.

**Primary outcome measures** Depression was assessed in 2009/10. Psychological distress, mental functioning, life satisfaction and self-rated health were assessed in 2009/10 and 2013/14.

**Results** Data were analysed using linear and logistic regression with adjustment for age, sex, household income, education, ethnicity and impairment category. Perceived disability discrimination was reported by 117 (13.4%) participants. Cross-sectionally, discrimination was associated with depression (OR=5.40, 95% CI 3.25 to 8.97) fair/poor self-rated health (OR=2.05; 95% CI 1.19 to 3.51), greater psychological distress (B=3.28, 95% CI 2.41 to 4.14), poorer mental functioning (B=−7.35; 95% CI −9.70 to −5.02) and life satisfaction (B=−1.27, 95% CI −1.66 to −0.87). Prospectively, discrimination was associated with increased psychological distress (B=2.88, 95% CI 1.39 to 4.36) and poorer mental functioning (B=−5.12; 95% CI −8.91 to −1.34), adjusting for baseline scores.

**Conclusions** Perceived disability-related discrimination is linked with poorer well-being. These findings underscore the need for interventions to combat disability discrimination.

## Strengths and limitations of this study

► The use of the prospective United Kingdom Household Longitudinal Study allowed us to examine well-being over a 4-year period.
► The disability discrimination measure took into account several kinds of discriminatory behaviour and included multiple settings where perceived disability discrimination could be encountered.
► Our findings are based on perceptions of disability discrimination rather than objective encounters with disability discrimination.
► Disability discrimination was only assessed at one point in time, meaning our measure does not necessarily reflect pervasive discrimination.

live in poverty than non-disabled adults,[5] with knock-on effects regarding access to health and other services.[6]

These practical difficulties experienced by a person with a long-term impairment may be aggravated by and interact with discrimination in the person's environment. Disability discrimination has been defined as unwanted, exploitative or abusive conduct against disabled people which violates their dignity and security or creates an intimidating or offensive environment.[7] Although disability is a protected characteristic under equality legislation,[1] disability discrimination is perceived to be common. In a 2015 population survey of 27 718 adults from 26 European countries, 50% of participants reported disability discrimination to be widespread, a 4% increase from 2012 data. However, the reasons behind this increase are unclear due to the cross-sectional nature of the Eurobarometer data.[8]

Disability discrimination can happen in a variety of settings including on the street, in the workplace and in public venues such as shops or pubs.[9] Recent British data suggest that disabled people are more likely than non-disabled people to report feeling unsafe when walking alone and to worry about physical attack and theft.[10 11] Disability is the second

## INTRODUCTION

'Disability' is an umbrella term for long-term impairments, activity limitations and participation restrictions, experienced by an individual with a health condition in interaction with their environment.[1–3] In the UK, the prevalence of self-reported disability is rising, with 21% of the population reporting a disability in 2017–2018, an increase from 18% in 2007–2008.[4] This increase is likely driven by population ageing. The majority of disabled people report experiencing difficulties in everyday life. For example, disabled people in Britain have lower educational attainment, are less likely to be employed and are more likely to

most common motivator for hate crime incidents, after race in England and Wales.[12] In the workplace, several studies suggest that a greater proportion of disabled than non-disabled individuals report experiences of discrimination.[7 13]

A growing body of research has investigated discrimination as a determinant of well-being.[14–16] Meta-analyses have linked perceived discrimination with depression and psychological distress and with poorer life satisfaction and self-rated health.[14–16] However, disability discrimination was not assessed in two of these meta-analyses and the majority of previous working has focused on racism.[14 15] In the most recent pooled analysis of 328 studies, of which only 8 studies concerned physical illness discrimination and disability discrimination,[16] the combined category of physical illness/disability discrimination was associated with greater psychological distress and lower self-esteem.

Several cross-sectional studies have assessed perceived disability discrimination alone in relation to well-being outcomes. One study of 229 individuals with an intellectual disability in England found that self-reported stigma was associated with a greater number of depression and anxiety symptoms.[17] Two analyses of the Australian Survey of Disability, Ageing and Carers (n=9655 and n=6183, respectively) linked disability discrimination with greater psychological distress.[18 19] This association was similarly observed in a Swedish general population survey.[20] Research has also linked perceived disability discrimination with lower life satisfaction in Canadian adolescents,[21] Korean women with severe disabilities[22] and Israeli nationals with physical disabilities.[23] Perceived disability discrimination has been associated with poorer self-rated health cross-sectionally in four studies[19 24–26] including a general population analysis of 52 458 individuals, from the European Social Survey.[25]

Cross-sectional correlations are difficult to interpret: perception of discrimination may result in emotional distress, but it is also possible that emotional distress leads to alterations in how people interpret social interactions with others. To date, only one study has assessed prospective associations between perceived disability discrimination and well-being.[27] In an analysis of older adults (≥50 years) participating in the US-based Health and Retirement Study (HRS), perceived disability discrimination was associated with poorer life satisfaction, self-rated health and greater loneliness over 4 year follow-up.

Overall, previous research has been dominated by cross-sectional studies, precluding the assessment of the temporal relationship between perceived disability discrimination and well-being outcomes. No longitudinal studies have compared people with a disability who do or do not report discrimination and well-being outcomes. To address these limitations, this study aimed to investigate cross-sectional and prospective associations between perceived disability discrimination and well-being in a UK population cohort.

## METHODS
### Study population
This study uses data from Understanding Society: The UK Household Longitudinal Study (UKHLS).[28] The overarching purpose of UKHLS is to provide high-quality longitudinal data about the health, work, education, income, family and social life of the UK population.[29] Data collection began in 2009/10 (wave 1) with follow-ups annually. The current study uses data from waves 1 (2009/10) and 5 (2013/14). These data were collected through face-to-face interview via computer-aided personal interview and self-completion paper questionnaires and from wave 3 via computer administered self-interview. The UKHLS comprises a representative general population probability sample of UK households, in addition to an ethnic minority boost sample.[29 30] The general population sample is based on proportionality stratified clustered samples of residential addresses in England, Scotland and Wales. In Northern Ireland, an unclustered systematic random sample of domestic addresses was selected. The ethnic minority boost was selected from high concentration ethnic minority areas, where 80% of the UK's five major ethnic minorities live.[29]

Our data come from the 'extra 5 min sample' of over 8000 participants who had an additional 5 min of questions on issues pertinent to ethnicity research including discrimination. This sample comprises mostly ethnic minorities (n=6722) who were drawn from the ethnic minority boost along with a comparison group of white participants (n=1428).[30] We restricted our sample to those who responded to the disability discrimination questions (n=4788) with a self-reported disability (n=871). Self-reported disability was based on a positive response the question "*Do you have any health problems or disabilities which mean you have substantial difficulties with any of the following areas of your life*" across any of the 12 types of difficulty assessed. These included issues with manual dexterity and mobility, problems with memory or the ability to concentrate, difficulties with learning and understanding as well as hearing and sight impairments. The response rates for the UKHLS general population sample and the ethnic minority boost at wave 1 were 81.8% and 72.4%, respectively.[30] The response rate for the 'extra 5 min sample' was 42.5%. At wave 5, there was loss to follow-up (n=431), leaving a follow-up sample size of 440 participants. Our definition of disability did not include mental health-related impairments.

### Perceived disability discrimination
To assess discrimination, participants were asked whether in the past 12 months they (a) felt unsafe, (b) avoided going to or being in, (c) had been insulted, called names, threatened or shouted at or (d) had been physically attacked, in seven different settings: 1) at school/college/work, 2) on public transport, 3) at or around bus or train stations, 4) in a taxi, 5) public buildings such as shopping centres or pubs, 6) outside on the street, in parks or other public places or 7) at home. If they answered

yes to any one of these questions, a follow-up question asked them to choose a reason from a list of categories including disability, sex and ethnicity among others. It was possible to choose multiple settings and attributions for the perceived discrimination. Those who attributed any experience of discrimination to disability are treated as cases of perceived disability discrimination.

## Outcome variables

Self-reported doctor-diagnosed clinical depression was measured at wave 1 (2009/10) with responses coded as yes/no. Depression was not analysed longitudinally due to a lack of incident cases. All other outcomes were assessed at waves 1 (2009/10) and 5 (2013/14). Psychological distress was measured using the General Health Questionnaire (GHQ)-12,[31] which involved ratings of 12 statements including whether the individual had "*Been able to enjoy your normal day to day activities*" or whether they '*felt constantly under strain*' with response options of 0='no' and 1='yes'. Total scores range from 0 (least distressed) to 12 (most distressed). The 12-item Short-Form Health Survey (SF-12) mental component summary score was used to measure limitations caused by emotional, mental health and social functioning issues.[32] Items included ratings of feelings experienced over the past 4 weeks such as "*Have you felt downhearted or blue?*" or "*Accomplished less than you would like*". Overall scores were derived using standard methods ranging from 0 (low functioning) to 100 (high functioning).[33] Life satisfaction was assessed using one item asking participants how satisfied they were with their 'life overall', with scores ranging from 1 (completely dissatisfied) to 7 (completely satisfied). Self-rated health was assessed using a single item: "*Would you say your health is…poor/fair/good/very good/excellent?*" In keeping with previous investigations,[34 35] self-rated health was dichotomised with 0 being 'good/very good/excellent' and 1 meaning 'poor/fair'.

## Covariates

A number of covariates (assessed at wave 1) that are likely relevant to perceived disability discrimination and well-being were selected a priori for inclusion in our analyses. Age in years was entered as a continuous variable, as there may be age differences in reports of discrimination[36] and in well-being outcomes.[37] Sex was included as a binary variable (male/female) based on previous literature demonstrating sex differences in the impact of discrimination on health.[38] Income and education were included as covariates as there may be socioeconomic differences in the perception of discrimination and in well-being outcomes.[36 39] Equivalised monthly household income was calculated by dividing total household net income by the modified Organization for Economic Cooperation and Development equivalence scale to adjust for the effects of household size and composition.[40] Income was entered as a continuous variable in our models. Education was included as a 3-level categorical variable: 1 'university degree', 2 'high school qualification' and 3

'no qualification'. As our sample was ethnically diverse, we included ethnicity as a 4-level variable with 1 being 'white' including those of white British, white Irish and any other white background, 2 being 'south Asian' including Indian, Pakistani and Bangladeshi individuals, 3 being 'black' including black African and black Caribbean participants and 4 being 'other' including individuals from Chinese and mixed backgrounds. There were four categories of impairment measured in the study: 'physical' disability which included difficulties with manual dexterity and mobility; 'cognitive' disability including problems with memory or the ability to concentrate, learn and understand; 'sensory' disability including hearing (apart from using a standard hearing aid) and sight impairments (apart from wearing standard glasses) and 'other' which encompassed reports of unspecified disability not captured in the other categories.

## Statistical analyses

We compared the characteristics of those who did and those who did not report disability discrimination at wave 1 using $\chi^2$ tests for categorical variables and independent samples t-tests for continuous variables. Associations between perceived disability discrimination and the various well-being measures were assessed using linear regression for continuous outcomes and logistic regression for categorical outcomes. For cross-sectional analyses, depression, psychological distress (GHQ-12), SF-12 mental component score, life satisfaction and self-rated health at wave 1 (2009/10) were the outcome variables. For prospective analyses, psychological distress (GHQ-12), SF-12 mental component score, life satisfaction and self-rated health at wave 5 (2013/14) were the outcomes. Age, sex, household income, education, ethnicity and disability type at wave 1 were controlled for in all analyses. Baseline (wave 1) scores/status on the relevant well-being variable was included as an additional covariate in prospective analyses. Only those with complete case information at wave 1 (n=871) and wave 5 (n=440) were included in the analyses. Results from linear regression analyses are presented as unstandardised B and 95% CIs. Results from logistic regression analyses are presented as ORs and 95% CI. All analyses were unweighted and conducted using SPSS V.24.

## Sensitivity analyses

We carried out three sensitivity analyses. In our first, we assessed whether those who were lost to follow-up (n=431) differed from those who provided data at both waves (n=440). We tested whether this impacted the results by conducting the cross-sectional analyses (wave 1) including only those who provided follow-up data at wave 5. We carried out our second sensitivity analysis to test the possibility that one of the four types discriminatory behaviour contributing to the measure of perceived disability discrimination (ie, feeling unsafe, avoiding somewhere, being insulted or being physically attacked) was driving the results. We tested this cross-sectionally

and prospectively by repeating our analyses removing each type of discriminatory behaviour in turn. In our third sensitivity analysis, we tested whether the prospective results from our complete case analysis at wave 5 (n=440) were similar when missing outcome information was imputed for those participants lost to follow-up (n=431).

## RESULTS

A total of 871 participants were included in the study and of these 117 (13.4%) reported perceived disability discrimination. Disability discrimination was the most commonly reported form of discrimination in the sample, followed by age discrimination (4.3%), sex discrimination (3.9%), ethnicity discrimination (3.8%), religious discrimination (2.2%) and discrimination on the basis of sexual orientation (0.5%). Of the categories of disability discrimination assessed, the most commonly reported was feeling unsafe (86.1%; 95% CI 79.48 to 92.74), followed by avoiding somewhere (72.8%; 95% CI 64.08 to 81.55), being insulted (23.5%; 95% CI 14.33 to 32.73) and being physically attacked (2.8%; 95% CI 0.04 to 5.98). The most common settings in which disability discrimination was reported were on the street (77.8%; 95% CI 70.13 to 85.42), in public buildings such as shops or pubs (59.8%; 95% CI 50.81 to 68.84), on public transport (51.3%; 95% CI 42.09 to 60.47) and at or around bus or train stations (40.2%; 95% CI 31.16 to 49.19). A quarter of participants reported experiencing disability discrimination at home (25%; 95% CI 17.61 to 33.67). Perceived disability discrimination was less frequently reported in school or workplace settings (12.8%; 95% CI 6.67 to 18.97) or in taxis (12%; 95% CI 6 to 17.93). The prevalence of the various types of perceived disability discrimination and the settings in which the discrimination occurred for different types of disability can be found in online supplementary table 1. There were no statistically significant differences between people with different types of disability in discrimination type or discrimination setting.

The baseline characteristics of the sample are displayed in table 1. The group who reported disability discrimination were younger on average (48.29±14.89 years) than those who did not report discrimination (53.42±16.56 years). They were more likely to be white (27.4% vs 20.3%) and to be better educated than those who did not report discrimination, with a greater proportion holding university degrees (28.2% vs 22.7%). Physical disability was most common in those who did not perceive discrimination (46.6%), whereas other unspecified disabilities (65%) were most frequently reported by those who perceived discrimination.

### Cross-sectional associations between perceived disability discrimination and well-being

Our findings suggest that individuals who perceived disability discrimination were significantly more likely

to report a diagnosis of clinical depression (OR=5.40; 95% CI 3.25 to 8.97, p<0.001) and were more likely to rate their health as fair/poor (OR=2.05; 95% CI 1.19 to 3.51, p=0.009) than those who did not perceive disability discrimination, independent of covariates (first panel table 2). Those who reported discrimination also had significantly higher levels of psychological distress (B=3.28, 95% CI 2.41 to 4.14, p<0.001), poorer mental functioning on the SF-12 (B=−7.35; 95% CI −9.70 to −5.02, p<0.001) and lower life satisfaction (B=−1.27, 95% CI −1.66 to −0.87, p<0.001), than those who did not report discrimination.

### Prospective associations between perceived disability discrimination and well-being

In prospective analyses (second panel table 2), those who reported perceived disability discrimination at wave 1 had higher levels of psychological distress 4 years later at wave 5 than those who did not report discrimination, independent of covariates and baseline psychological distress (B=2.88, 95% CI 1.39 to 4.36, p<0.001). We detected a prospective association between perceived disability discrimination at wave 1 and poorer SF-12 mental functioning at wave 5 (B=−5.12; 95% CI −8.91 to −1.34, p=0.008). Those who reported disability discrimination at wave 1 had slightly lower life satisfaction (means=4.14 vs 4.67) and a greater proportion rated their health as fair/poor (67.3% vs 62.1%) than those who did not report discrimination at follow-up (wave 5). However, these differences did not reach statistical significance.

### Sensitivity analyses

In the first sensitivity analysis (table 3), cross-sectional findings for those who provided complete data at wave 5 were similar to the full-sample at wave 1. The demographic characteristics of those lost to follow-up were similar to those of complete cases (table 4). Only education differed significantly between the groups, with those who provided complete data at wave 5 more likely to hold a degree (27.0%) than those lost to follow-up (19.7%).

In the second sensitivity analysis, removing each of the discriminatory behaviours from the measure of discrimination in turn did not alter the cross-sectional results (table 5). Prospectively, the association between perceived disability discrimination and increased psychological distress remained the same regardless of the type of discriminatory behaviour removed from the measure. For SF-12 mental functioning, the association was fairly robust to the type of discriminatory behaviour, but was slightly attenuated when 'feeling unsafe' was removed from the discrimination variable (p=0.058). Again, no significant prospective associations were detected for life satisfaction and self-rated health.

In our final sensitivity analysis (online supplementary table 2), we repeated the prospective analyses with imputation for missing outcome information of

**Table 1** Associations between perceived disability discrimination and sociodemographic factors (wave 1) and well-being measures (waves 1 and 5)

| | Overall sample (n=871) | No perceived discrimination (n=754) | Perceived discrimination (n=117) | P value |
|---|---|---|---|---|
| Age (years) | 52.73 (16.43) | 53.42 (16.56) | 48.29 (14.89) | 0.001 |
| 17–34 | 120 (13.7%) | 100 (13.3%) | 20 (17.1%) | |
| 35–44 | 172 (19.7%) | 149 (19.8%) | 23 (19.7%) | |
| 45–54 | 184 (21.1%) | 150 (19.9%) | 34 (29.1%) | |
| 55–64 | 157 (18.0%) | 134 (17.8%) | 23 (19.7%) | |
| 65+ | 238 (27.3%) | 221 (29.3%) | 17 (14.5%) | |
| Sex (% men) | 388 (44.5%) | 334 (44.3%) | 54 (46.2%) | 0.707 |
| Household income (£) | 1118.42 (902.54) | 1123.28 (930.47) | 1087.07 (698.65) | 0.687 |
| 0–499 | 117 (13.4%) | 105 (13.9%) | 12 (10.3%) | |
| 500–999 | 369 (42.4%) | 314 (41.6%) | 55 (47.0%) | |
| 1000–1499 | 216 (24.8%) | 189 (25.1%) | 27 (23.1%) | |
| 1500–1999 | 90 (10.3%) | 77 (10.2%) | 13 (11.1%) | |
| 2000+ | 79 (9.1%) | 69 (9.2%) | 10 (8.5%) | |
| Education (% yes) | | | | 0.003 |
| University degree | 204 (23.4%) | 171 (22.7%) | 33 (28.2%) | – |
| School qualification | 342 (39.3%) | 285 (37.8%) | 57 (48.7%) | – |
| No qualification | 325 (37.3%) | 298 (39.5%) | 27 (23.1%) | – |
| Ethnicity | | | | 0.002 |
| White | 185 (21.2%) | 153 (20.3%) | 32 (27.4%) | – |
| South Asian | 369 (42.4%) | 333 (44.2%) | 36 (30.8%) | – |
| Black | 189 (21.7%) | 168 (22.3%) | 21 (17.9%) | – |
| Other | 128 (14.7%) | 100 (13.3%) | 28 (23.9%) | – |
| Disability type | | | | <0.001 |
| Physical | 367 (42.1%) | 351 (46.6%) | 16 (13.7%) | |
| Sensory | 95 (10.9%) | 84 (11.1%) | 11 (9.4%) | |
| Cognitive | 87 (10.0%) | 73 (9.7%) | 14 (12.0%) | |
| Other | 322 (37.0%) | 246 (32.6%) | 76 (65.0%) | |
| Well-being measures (wave 1) | | | | |
| Depression | 115 (13.2%) | 65 (8.7%) | 50 (42.7%) | <0.001 |
| Psychological distress | 3.43 (3.85) | 2.85 (3.49) | 6.65 (4.18) | <0.001 |
| SF-12 mental | 43.87 (12.61) | 45.16 (12.22) | 35.72 (12.02) | <0.001 |
| Life satisfaction | 4.49 (1.71) | 4.71 (1.63) | 3.29 (1.67) | <0.001 |
| Fair/Poor self-rated health | 620 (71.2%) | 525 (69.6%) | 95 (81.2%) | 0.010 |
| Well-being measures (wave 5) | | | | |
| Psychological distress | 3.33 (3.99) | 2.86 (3.82) | 5.98 (3.91) | <0.001 |
| SF-12 mental | 44.38 (12.20) | 45.79 (11.62) | 36.47 (12.51) | <0.001 |
| Life satisfaction | 4.48 (1.61) | 4.62 (1.57) | 3.67 (1.64) | <0.001 |
| Fair/Poor self-rated health | 276 (62.7%) | 235 (61.0%) | 41 (74.5%) | 0.053 |

Data are presented as means (SD) and n (%). Percentages are valid per cent.
SF-12, 12-Item Short-Form Health Survey.

participants lost to follow-up (n=431). The prospective relationship between perceived disability discrimination and poor SF-12 mental functioning remained (p=0.034). However, there was no longer a statistically significant prospective association between reported discrimination and psychological distress (p=0.128).

**Table 2** Cross-sectional and prospective associations between perceived disability discrimination and well-being outcomes

| | Depression | Psychological distress | SF-12 mental | Life satisfaction | Fair/Poor self-rated health |
|---|---|---|---|---|---|
| | OR (95% CI) | B (95% CI) | B (95% CI) | B (95% CI) | B (95% CI) |
| **Wave 1** | | | | | |
| No perceived discrimination | 1 (reference)† | Reference‡ | Reference§ | Reference¶ | 1 (reference)†† |
| Perceived discrimination | 5.40 (3.25 to 8.97)*** | 3.28 (2.41 to 4.14)*** | −7.35 (−9.70 to −5.02)*** | −1.27 (−1.66 to −0.87)*** | 2.05 (1.19 to 3.51)** |
| **Wave 5** | | | | | |
| No perceived discrimination | – | Reference‡‡ | Reference§§ | Reference¶¶ | 1 (reference)††† |
| Perceived discrimination | – | 2.88 (1.39 to 4.36)*** | −5.12 (−8.91 to −1.34)** | −0.53 (−1.18 to 0.11) | 1.29 (0.59 to 2.83) |

All analyses are adjusted for age, sex, household income, education, ethnicity and disability type. Prospective analyses are additionally adjusted for baseline well-being status/score.
Possible scores on the psychological distress scale range from 0 to 12, SF-12 mental component scale range from 0 to 100 and the life satisfaction scale scores range from 0 to 7.
*p<0.05.
† n= 751 for the no perceived discrimination group; n= 117 for the perceived discrimination group.
‡ n= 454 for the no perceived discrimination group; n= 82 for the perceived discrimination group.
§ n= 742 for the no perceived discrimination group; n= 117 for the perceived discrimination group.
¶ n= 454 for the no perceiveddiscrimination group; n= 84 for the perceived discrimination group.
**p<0.01.
††n= 754 for the no perceived discrimination group; n= 117 for the perceived discrimination group.
‡‡ n= 177 for the no perceived discrimination group; n= 31 for the perceived discrimination group.
§§ n= 239 for the no perceived discrimination group; n= 43 for the perceived discrimination group.
¶¶ n= 171 for the no perceived discrimination group; n= 34 for the perceived discrimination group
***p <0.001.
††† n= 385 for the no perceived discrimination group; n= 55 for the perceived discrimination group.
SF-12, 12-Item Short-Form Health Survey.

## DISCUSSION

In a sample of UK-based participants with self-reported disability, perceived discrimination was associated with higher prevalence of depression, greater psychological distress and poorer mental functioning, life satisfaction and self-rated health. Prospectively, disability discrimination was associated with increased psychological distress and worse mental functioning 4 years later. Our results were robust to adjustment for a range of covariates and were not driven by any specific kind of discriminatory behaviour. No significant prospective relationships with life satisfaction and self-rated health were observed.

Previous literature has been dominated by cross-sectional studies. To our knowledge, only one previous study has investigated the prospective association between disability discrimination and well-being outcomes. In this analysis of US adults from the HRS cohort, perceived disability discrimination was associated with poorer life satisfaction and self-rated health over 4-year follow-up.[27] In the current study, we observed poorer mental functioning and greater psychological distress 4 years later in those who reported disability discrimination, taking into account baseline scores on these variables. We failed to detect a significant association between perceived discrimination and life satisfaction or self-rated health at follow-up. Although on average, those who perceived disability discrimination in our sample had poorer life satisfaction and were more likely to rate their health as fair/poor at follow-up than those who did not perceive discrimination, these differences did not reach statistical significance. One reason for the divergence in findings between our study and the HRS analysis[27] may be study design. We limited our analyses to those with self-reported disability, whereas in the HRS study associations between well-being and disability discrimination were assessed across the entire sample. Our analysis offers more precision in the assessment of the relationship between disability discrimination and well-being outcomes, by directly comparing people with disability who did and did not perceive discrimination. Another possibility for these null findings may be that significant associations between discrimination and life satisfaction and self-rated health do not become apparent until older adulthood, perhaps allowing for repeated exposures to disability discrimination. However, this assertion remains to be tested. Another potential explanation is that the impact of ongoing disability discrimination on life satisfaction and self-rated health in our sample had already become apparent at the time of the baseline survey, limiting the scope for further decline.

Our study adds to the cross-sectional literature linking perceived disability discrimination and poorer well-being outcomes by demonstrating associations in a community sample of disabled people living in the UK. Our results extend the findings of an earlier study linking stigma and depression in those with intellectual disability,[17] by

**Table 3** Cross-sectional and prospective associations between perceived disability discrimination and well-being outcomes (complete cases at wave 5)

| | Psychological distress† | SF-12 mental‡ | Life satisfaction§ | Fair/Poor self-rated health¶ |
| --- | --- | --- | --- | --- |
| | B (95% CI) | B (95% CI) | B (95% CI) | B (95% CI) |
| **Wave 1** | | | | |
| No perceived discrimination | Reference | Reference | Reference | 1 (reference) |
| Perceived discrimination | 2.65 (1.21 to 4.08)*** | −7.20 (−11.01 to −3.39)*** | −1.27 (−1.91 to −0.63)*** | 2.66 (1.16 to 6.08)* |
| **Wave 5** | | | | |
| No perceived discrimination | Reference | Reference | Reference | 1 (reference) |
| Perceived discrimination | 2.88 (1.39 to 4.36)*** | −5.12 (−8.91 to −1.34)** | −0.53 (−1.18 to 0.11) | 1.29 (0.59 to 2.83) |

All analyses are adjusted for age, sex, household income, education, ethnicity and disability type. Prospective analyses are additionally adjusted for baseline well-being status/score.

Possible scores on the psychological distress scale range from 0 to 12, SF-12 mental component scale range from 0 to 100 and the life satisfaction scale scores range from 0 to 7.

*p<0.05.

†n= 177 for the no perceived discrimination group; n= 31 for the perceived discrimination group.

‡n= 239 for the no perceived discrimination group; n= 43 for the perceived discrimination group.

§n= 171 for the no perceived discrimination group; n= 34 for the perceived discrimination group.

¶n= 385 for the no perceived discrimination group; n= 55 for the perceived discrimination group.

**p<0.01, ***p <0.001

SF-12, 12-Item Short-Form Health Survey.

establishing this relationship in a sample with a broader range of disability. In keeping with previous studies, we observed greater psychological distress[18–20] and poorer life satisfaction[21–23 27] in those who reported disability discrimination. Our study adds to this existing evidence by demonstrating this link in a UK-based sample for the first time. Similar to earlier work from Australian, European and North American samples,[19 24 27 41] we observed a relationship between perceived disability discrimination and poorer self-rated health. Cross-sectional studies cannot determine whether perceived disability discrimination predicts poor mental well-being, or whether perceptions of discrimination are an indicator of psychological distress. Our prospective findings therefore add to the field in establishing that perceived disability discrimination predicts psychological distress and poorer mental functioning, net of baseline associations, so has negative implications for future well-being.

This is an observational study and longitudinal analyses do not necessarily imply causality. There could be unmeasured factors responsible for the associations that emerged. Nevertheless, with regard to the pathways linking perceived disability discrimination and well-being, there are several possibilities that could explain our results. One mechanism could be that perceptions of disability discrimination in healthcare settings serve to impede access to health services. An analysis of HRS found that reports of frequent discrimination in healthcare settings were predictive of new or worsened disability over 4-year follow-up.[42] Quantitative[38 43] and qualitative[44] evidence

suggests that those who perceive disability discrimination are less likely to seek healthcare. However, there may be sex differences in this association, with a Swedish study only detecting a relationship between disability discrimination and healthcare avoidance in women.[38] However, no interaction between sex and perceived discrimination was detected in the current study (data not shown).

Poor health behaviours are another potential mechanism linking disability discrimination and poorer well-being. For example, perceived disability discrimination has been linked with worse sleep quality in the HRS, with psychological distress acting as a full mediator of this association.[45] It is possible that disabled people could engage in negative health behaviours as a means of coping with the psychological impact of discrimination. In a study of 304 individuals with disability, perceived disability discrimination was positively associated with illicit drug use.[46] Eating may offer a source of comfort in the face of discrimination.[47] A US study of over 5000 individuals observed a link between physical disability discrimination and overeating.[48]

Another possibility is that perceived disability discrimination and well-being are linked through disturbed stress-related biological processes. In line with the theory of allostatic load, perceived chronic discrimination causing frequent activation of the stress response system, could over time result in disturbances across multiple biological systems.[49] Systematic reviews and meta-analyses, which have predominately focused on racism, suggest that discrimination is linked with heightened cardiovascular

**Table 4** Participant characteristics at wave 1 (2009/10) of complete cases and those lost to follow-up

| | Lost to follow-up (n=431) | Complete cases (n=440) | P value |
|---|---|---|---|
| Age (years) | 52.26 (17.08) | 53.19 (15.76) | 0.407 |
| 17–34 | 25 (5.8%) | 12 (2.7%) | |
| 35–44 | 46 (10.7%) | 37 (8.4%) | |
| 45–54 | 79 (18.4%) | 93 (21.1%) | |
| 55–64 | 91 (21.2%) | 93 (21.1%) | |
| 65+ | 189 (44.0%) | 205 (46.6%) | |
| Sex (% men) | 200 (46.4%) | 188 (42.7%) | 0.275 |
| Household income (£) | 1101.26 (1037.11) | 1135.22 (748.46) | 0.579 |
| 0–499 | 69 (16.0%) | 48 (10.9%) | |
| 500–999 | 175 (40.6%) | 194 (44.1%) | |
| 1000–1499 | 113 (26.2%) | 103 (23.4%) | |
| 1500–1999 | 42 (9.7%) | 48 (10.9%) | |
| 2000+ | 32 (7.4%) | 47 (10.7%) | |
| Education (% yes) | | | 0.024 |
| University degree | 85 (19.7%) | 119 (27.0%) | – |
| School qualification | 171 (39.7%) | 171 (38.9%) | – |
| No qualification | 175 (40.6%) | 150 (34.1%) | – |
| Ethnicity | | | 0.213 |
| White | 88 (20.4%) | 97 (22.0%) | – |
| South Asian | 172 (39.9%) | 197 (44.8%) | – |
| Black | 105 (24.4%) | 84 (19.1%) | – |
| Other | 66 (15.3%) | 62 (14.1%) | – |
| Disability type | | | 0.189 |
| Physical | 166 (38.5%) | 201 (45.7%) | |
| Sensory | 51 (11.8%) | 44 (10.0%) | |
| Cognitive | 47 (10.9%) | 40 (9.1%) | |
| Other | 167 (38.7%) | 155 (35.2%) | |
| Mental well-being | | | |
| Psychological distress | 3.46 (3.85) | 3.41 (3.86) | 0.874 |
| SF-12 | 43.72 (12.93) | 44.01 (12.30) | 0.741 |
| Life satisfaction | 4.40 (1.75) | 4.56 (1.69) | 0.306 |
| Self-rated health (% fair/poor) | 319 (74.0%) | 301 (68.4%) | 0.068 |

Data are presented as means (SD) and n (%).
*Complete cases are defined as those who were present at wave 1 and provided data on at least one well-being measure at wave 5.
SF-12, 12-Item Short-Form Health Survey.

stress reactivity,[14 50] while race[50–52] and weight discrimination[53] have been linked with alterations in cortisol. To our knowledge, no study has investigated associations between perceived disability discrimination and changes in cardiovascular or neuroendocrine activity. In the HRS cohort, perceived disability discrimination was linked with raised C reactive protein levels cross-sectionally.[54] Heightened inflammation is thought to be predictive of poorer mental well-being,[55] offering a plausible pathway between perceived disability discrimination and later psychological distress and poorer mental functioning seen in the present study. Further work is required to confirm this

assertion, particularly as the HRS analysis was not limited to those with a confirmed disability.

Our study had several strengths. The use of the UKHLS cohort allowed us to examine well-being over a 4-year period across a wide age range (17–96 years), while adjusting statistically for factors that could confound associations. The discrimination measure took into account several kinds of discriminatory behaviour and included multiple settings where perceived disability discrimination could be encountered.

However, the study was not without limitations. We lost a considerable number of participants at follow-up,

**Table 5** Sensitivity analysis: perceived disability discrimination measure excluding each discriminatory behaviour in turn

| Cross-sectional analyses (wave 1) | | Model 1 | Model 2 | Model 3 | Model 4 |
|---|---|---|---|---|---|
| Depression | OR (95% CI) | 4.41 (2.55 to 7.60)*** | 5.24 (3.06 to 8.98)*** | 4.80 (2.92 to 7.88)*** | 5.41 (3.26 to 8.98)*** |
| Psychological distress | Coeff. (95% CI) | 3.64 (2.68 to 4.60)*** | 3.28 (2.33 to 4.23)*** | 3.13 (2.26 to 4.01)*** | 3.27 (2.41 to 4.14)*** |
| SF-12 mental | Coeff. (95% CI) | −6.63 (−9.29 to −3.97)*** | −7.61 (−10.15 to −5.08)*** | −7.53 (−9.89 to −5.18)*** | −7.35 (−9.68 to −5.02)*** |
| Life satisfaction | Coeff. (95% CI) | −1.23 (−1.69 to −0.76)*** | −1.45 (−1.88 to −1.02)*** | −1.24 (−1.64 to −0.85)*** | −1.27 (−1.66 to −0.87)*** |
| Fair/Poor self-rated health | OR (95% CI) | 1.92 (1.04 to 3.53)* | 2.39 (1.32 to 4.33)** | 2.21 (1.28 to 3.81)** | 2.04 (1.19 to 3.50)** |
| **Prospective analyses (wave 5)** | | **Model 1** | **Model 2** | **Model 3** | **Model 4** |
| Psychological distress | Coeff. (95% CI) | 2.78 (1.14 to 4.41)*** | 2.69 (1.15 to 4.24)*** | 2.89 (1.45 to 4.33)*** | 2.88 (1.39 to 4.36)*** |
| SF-12 mental | Coeff. (95% CI) | −4.33 (−8.81 to 0.14) | −4.90 (−8.86 to −0.94)** | −5.94 (−9.59 to −2.28)** | −5.13 (−8.91 to −1.34)** |
| Life satisfaction | Coeff. (95% CI) | −0.39 (−1.14 to 0.36) | −0.37 (−1.06 to 0.32) | −0.39 (−1.02 to 0.25) | −0.53 (−1.18 to 0.11) |
| Fair/Poor self-rated health | OR (95% CI) | 1.23 (0.50 to 3.03) | 1.30 (0.57 to 2.95) | 1.31 (0.61 to 2.83) | 1.29 (0.59 to 2.83) |

All analyses are adjusted for age, sex, household income, education, ethnicity and disability type. Prospective analyses are additionally adjusted for baseline well-being status/score.
Model 1 excludes 'felt unsafe at some place' from the measure of perceived disability discrimination; model 2 excludes 'avoided some place'; model 3 excludes 'was insulted at some place' and model 4 excludes 'was attacked at some place'.
Possible scores on the psychological distress scale range from 0 to 12, SF-12 mental component scale range from 0 to 100 and the life satisfaction scale scores range from 0 to 7.
*P<0.05, **p<0.01, ***p<0.001.
Coeff, unstandardised B coefficient.

and although the cross-sectional findings did not differ between those who provided data at both waves and those lost to follow-up (sensitivity analysis 1), we cannot be sure that selection bias due to low retention did not impact our findings longitudinally due to the extent of missing data. Indeed, in imputed analyses (sensitivity analysis 3) the association between disability discrimination and well-being held for SF-12 mental functioning but not for psychological distress. Our findings are based on perceptions of disability discrimination rather than objective encounters with disability discrimination. It is possible that perceiving oneself as a target for discrimination and objective encounters with discrimination could have differing consequences for well-being. Indeed, earlier work in a sample with significant health limitations indicates that individuals with poorer mental well-being may be more likely to perceive stigma.[56] Future studies assessing reciprocal prospective associations between perceived disability discrimination and well-being could help to clarify this issue. Our discrimination measure was based on self-reports of experiences during the past 12 months and was therefore subject to recall bias. Furthermore, this measure was not specific to disability discrimination. The fact that participants were able to attribute multiple reasons for their experience of discrimination, could have helped avoid priming or bias. Other tools specifically designed to assess disability discrimination could have garnered different results. Our sample was ethnically diverse, and we took ethnicity into account in our models. Although disability discrimination was

the most commonly reported form of discrimination in this sample, perceived discrimination on the basis of ethnicity may also have been relevant for this sample. Further work is required to understand how disability discrimination interacts with ethnicity discrimination, as well as other types of discrimination to influence well-being. Disability discrimination was only assessed at one point in time, meaning our measure does not necessarily reflect pervasive discrimination. However, other work in UKHLS suggests that perceived disability discrimination is still frequently reported at later stages of data collection.[57] Future research is required to determine whether perceptions of disability discrimination are persistent or alter over time. We operationalised perceived discrimination as a simple binary variable and had no information on the frequency of encounters with discrimination over time. Therefore, the potential dose-response relationship between the frequency of discrimination and well-being remains to be elucidated. Our study included participants with physical, cognitive and sensory disabilities. However, our sample is unlikely to have captured those with severe cognitive impairments due to the demands of survey participation. For a large proportion (37%), their disability type was unknown and classified as 'other', limiting our understanding. While, no one with a mental health-related disability was included in the physical, cognitive and sensory disabilities categories, we cannot be certain that the 'other' category did not include participants with mental health-related impairments.

Overall, our study adds to the literature by demonstrating prospective associations between perceived disability discrimination and well-being outcomes. These findings emphasise the need to reduce the prevalence of disability discrimination, with the benefit of promoting equality as well as possible advantages for well-being too. Although complete elimination of disability discrimination is likely to be difficult, recognition of disability discrimination as an issue is the first step in preventing its occurrence. Addressing this could involve raising awareness through the use of campaigns. The Public Sector Equality Duty in the UK requires public bodies to have due regard to the need to eliminate discrimination and this awareness raising should begin early in life.[7] However, it is estimated that <40% of English primary schools have a disability equality scheme in place, with race and gender equality more often prioritised over disability equality.[58] Therefore, further effort on this issue is required,[7] particularly as disability discrimination is perceived to be more widespread than gender discrimination in Europe.[8]

As well as macro-level awareness raising, on an individual basis the negative impact of perceived disability discrimination on well-being may be buffered through the use of social support. In two cross-sectional studies of US adults with varied disability diagnoses, those with more friends reported greater life satisfaction and these friendships attenuated the link between functional impairment and poorer quality of life.[59] In an Israeli study, perceived disability discrimination and poorer life satisfaction were only linked in those with low and moderate levels of social support, with no association in those with greater levels of support.[60] Further research on disability discrimination is necessary to develop awareness campaigns and to appropriately target individual-level interventions.

**Contributors** RAH conducted the statistical analysis and wrote the manuscript. AS edited and reviewed the manuscript. RPL edited and reviewed the manuscript. SEJ provided scientific overview, edited and reviewed the manuscript.

**Funding** This work was supported by the Economic and Social Research Council https://esrc.ukri.org/, grant number ES/R005990/1.

**Disclaimer** The funders had no role in study design, data collection and analysis, decision to publish or preparation of the manuscript.

**Competing interests** None declared.

**Patient and public involvement** Patients and/or the public were not involved in the design, conduct, reporting or dissemination plans of this research.

**Patient consent for publication** Not required.

**Ethics approval** Ethical approval for UKHLS was obtained from the University of Essex Ethics Committee.

**Provenance and peer review** Not commissioned; externally peer reviewed.

**Data availability statement** Data are available in a public, open access repository. The UKHLS datasets analysed during the current study are freely available in the UK Data Service repository https://ukdataservice.ac.uk/

**ORCID iDs**
Ruth A. Hackett http://orcid.org/0000-0002-5428-2950
Sarah E. Jackson http://orcid.org/0000-0001-5658-6168

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
