## [Reviewer comments · BMJ Open]

ARTICLE DETAILS

TITLE (PROVISIONAL)	Disability discrimination and wellbeing: A prospective analysis
AUTHORS	Hackett, Ruth; Steptoe, Andrew; Lang, Raymond; Jackson, Sarah

VERSION 1 - REVIEW

REVIEWER	Eric Emerson Lancaster University, UK
REVIEW RETURNED	09-Dec-2019

GENERAL COMMENTS	This is a well written paper that adds significantly to knowledge about the association between disability discrimination and health. My comments and suggestions all address relatively minor issues. 1. It is important to be clear throughout that the analytic sample is comprised of people with self-reported disability. So, for example, it is incorrect in the Introduction to assert on the basis of FRS data that 'the prevalence of disability is rising'. Better to insert 'self-reported' before disability. Similarly in the Discussion it is misleading to use the term 'confirmed disability'.2. The review of existing studies omits two cross sectional population based studies that have examined that association between disability discrimination and health: Alvarez-Galvez J. Measuring the effect of ethnic and non-ethnic discrimination on Europeans' self-rated health. International Journal of Public Health. 2016; 61:367–74; Emerson E. Self-reported exposure to disablism is associated with poorer self-reported health and wellbeing among adults with intellectual disabilities in England: cross sectional survey. Public Health. 2010; 124.3. In the Method section it would be useful to quote the specific items used to identify people with self-reported disability.4. Given that the association between age and disability is clearly non-linear, would not it be more appropriate to treat age as a categorical variable in the analyses? I assume that income was treated as a continuous variable and that education was treated as a categorical variable, but this should be stated.5. Were sample weights used in the analyses and did the analyses use the SPSS facility for complex sample design?6. The authors may be interested in the following paper that explored exposure to some of the discrimination variables in
---

	Understanding Society among participants with/without self-reported disability. Emerson, E., Krnjacki, K., Llewellyn, G., Vaughan, C., Kavanagh, A. (2016). Perceptions of safety and exposure to violence in public places among working age adults with disabilities or long-term health conditions in the UK: Cross sectional study. Public Health 135, 91-6.
--	--

REVIEWER	Daniel Pérez-Garín UNED
REVIEW RETURNED	12-Dec-2019

GENERAL COMMENTS	I think the present research is a relevant contribution to the literature about disability discrimination. As authors say, there are almost no works assessing the impact of discrimination with a longitudinal design. However, I think there are a number of questions that need to be addressed before the manuscript is accepted for publication. Methods Authors do not specify how the data from The United Kingdom Household Longitudinal Study were collected. Authors should explain why did they not decide to use a previously validated measure of discrimination. Covariates On line 145, authors say that their definition of disability did not include mental health-related impairments. However, I wonder whether they specifically excluded them from the “other” category that is mentioned on lines 197 and 198, and which types of disabilities were included in it. Since the population from which the sample was taken was not entirely comprised by people with disabilities, and many participants were members of several ethnic minorities, I think the effect of other types of discrimination, such as perceived racial discrimination should also be taken into account. Was this controlled for? Results I think the criterion used to categorize participants who perceived discrimination is unclear. There were several different questions. Was it enough to answer “yes” to any of them? Did they have to answer “yes” to at least half of them? Tables In the file I reviewed, tables 1 and 2 overlap, as do tables 4 and 5, thus, I could not visualize them correctly and was not able to review them.
---

REVIEWER	Christine Fekete Swiss Paraplegic Research, Nottwil, Switzerland University of Lucerne, Switzerland
REVIEW RETURNED	16-Dec-2019

GENERAL COMMENTS	Review of the manuscript entitled „Disability discrimination and mental health: A prospective analysis“ bmjopen-2019-035714 Thank you for giving me the opportunity to review the above mentioned manuscript. The study investigates cross-sectional as well as prospective associations of discrimination due to disability
--

and a range of health-related outcomes. The manuscript is clearly structured, well-written and touches a topic of high relevance. Please find my comments below.

- General: The outcomes of the study are rather diverse, ranging from measures assessing psychological distress (GHQ), general mental health (SF-12) and life satisfaction to self-reported (general) health. Terms to describe the 'overall outcome' of the study are inconsistently used ('mental health', 'mental wellbeing', 'mental health & wellbeing outcomes', 'emotional wellbeing' in the Tables...) and although concepts correlate to some extent, it seems problematic to use 'mental health' as overall term to encompass life satisfaction & self-reported health. Could the authors think of a rationale of why general health or life satisfaction are subsumed under mental health or adapt terminology to better capture the diversity of concepts?

Article summary

- The authors state that they examine 'changes in mental wellbeing'. However, change is not assessed with the prognostic model used. Please revise sentence accordingly.

Introduction

- I wonder whether the authors could link the concepts of disability to their argumentation in the second paragraph. I.e., 'these practical difficulties are mainly activity limitations and participation restrictions that might be aggravated by environmental factors, such as discrimination.'

- line 79f. Would be worthwhile to shortly discuss reasons for the increase in disability prevalence, is population ageing the major driver? Same for line 91f., would be interesting to have an idea why disability discrimination is rising, is there a change in social climate or because disability prevalence has risen?

- The meta-analytic review by Schmitt et al. seems to have included only 8 studies that included physical illness/disability discrimination. Suggestion to mention this low number as another argument for conducting this study (preventing that the vast number of 328 studies mentioned (line 104) leaves the reader with the question of why it is worthwhile to do another study on discrimination and mental health).

Methods

Study population:

- Please add some more details on the UKHLS study, i.e. what was the rationale for this study; what were its inclusion criteria; what was the sampling frame; What was the response rate of wave 1 and what was the response rate/drop-out rate at wave 5?

- Is the 'ethnic minority boost sample' the same as the 'extra 5 minutes sample'?

Perceived disability discrimination:

- Wonder whether the authors thought about using more fine graded information in order to better depict the frequency of discrimination (i.e. summing up the discrimination experiences in different setting). This would enable the investigation of a dose-response relationship between discrimination and mental health what would add to the value of the paper.

Covariates:

- line 182: 'covariates that are likely relevant': Would be good if the covariates included in the models would have been empirically tested for their association with the predictors and the outcomes in that sample. Including covariates that are not associated with both, the predictors and the outcomes reduce the precision of the estimates and could be removed from the models.

- line 197: It remains unclear what kind of impairments were classified into the category 'other' and given that around 40% of the sample belong to this category, it would be worthwhile thinking about a more fine-graded classification (if information is available). Table 1 shows for example that the majority of persons perceiving discrimination belong to the category 'other' (65%), it would thus be important to better characterize this group.

- Given that this sample consists of mostly ethnic minorities (line 141), the double burden of being part of an ethnic minority and having a disability is not accounted for in analysis. Please consider adjusting analysis for 'discrimination due to ethnicity' or critically discuss this issue.

Statistical analysis:

- Please specify the how missing values were treated. It seems that these are only full case analysis. It is strongly recommended to use multiple imputation to account for item-non-response.

- It is a pity that the predictor variable is dichotomized. Seeing a categorical variable summing up the discrimination experiences would enable to detect a kind of a dose-response relationship between discrimination and the health outcomes.

Results

- Tab 1: Basic description of outcome variables should be placed in Table 1 rather than in Table 2 and column with the characteristics of the total population would be helpful. Adding indicators of which wave data come from for outcome variables would help the reader.

- Tab 2: It is difficult to read the table as one is used to see the outcome variable in the top row instead of predictor variable; please put outcomes in the top row and the predictors in the first column; results of prospective analysis should add information on the wave at which data were collected.

- lines 263ff: Would help the reader to add information on the measurement time point, e.g. "...those who reported perceived disability discrimination at wave 1 had higher levels of" "Those who reported disability discrimination at wave 1 had slightly lower life satisfaction at wave 5..."

Discussion

- May discuss the fact that persons with severe cognitive impairments were not included in the study sample as the survey inclusion criteria was probably prerequisite for study participation was the ability to respond to the questions.

- lines 313ff: Another possibility for the null finding might also be that the measurement of perceived disability discrimination is rather crude in terms of frequency and burden. One can assume

	that people facing discrimination repeatedly and frequently are more affected in terms of health than someone who was exposed to a single situation of discrimination over one year... Therefore, a more fine-grained analysis using kind of a sumscore over all 4 discrimination variables might reveal different findings. - The interpretation of results in paragraphs 3 and 4 is interesting, thank you. Are there other studies showing a more immediate or direct effect of discrimination on affectivity? Experiencing discrimination is highly likely to increase negative affectivity, such as anger, sadness, and that repeated experiences may increase the risk of psychological distress or depression if adequate coping is not given? - Last paragraph: it is a bit unlucky to close the paper with this paragraph as it again contains a discussion on mechanisms underlying the observed association. Suggestion to place this (interesting) paragraph before the strength/limitations section and ending the article with concrete suggestions for individual-level interventions (one of which can then be 'strengthening social support').
--	--

REVIEWER	Anne Kavanagh University of Melbourne, Australia
REVIEW RETURNED	21-Dec-2019

GENERAL COMMENTS	This is a well written paper which makes an important contribution to the literature. To improve the manuscript I suggest the following amendments:  1. There is considerable loss to follow-up but the authors do not discuss this in the Discussion. The sensitivity analysis of the cross-sectional data is reassuring but does not ensure that selection bias due to low retention is not a major problem longitudinally. This should be commented on in the Discussion. 2. I was not clear why the second sensitivity analysis was conducted. 3. There was no rationale for testing of interactions. Unless these are incorporated into the research questions I suggest they are deleted.
---

VERSION 1 – AUTHOR RESPONSE

Reviewer 1: Eric Emerson

This is a well written paper that adds significantly to knowledge about the association between disability discrimination and health. My comments and suggestions all address relatively minor issues.

1. It is important to be clear throughout that the analytic sample is comprised of people with self-reported disability. So, for example, it is incorrect in the Introduction to assert on the basis of FRS data that 'the prevalence of disability is rising'. Better to insert 'self-reported' before disability. Similarly in the Discussion it is misleading to use the term 'confirmed disability'.

Thank you for taking the time to review this paper. This is a worthwhile suggestion. We have revised the paper to highlight that the sample comprised of people with self-reported disability in the Introduction (see page 4, line 79) and Discussion sections (see page 14, line 347).

2. The review of existing studies omits two cross sectional population based studies that have examined that association between disability discrimination and health: Alvarez-Galvez J. Measuring the effect of ethnic and non-ethnic discrimination on Europeans' self-rated health. *International Journal of Public Health*. 2016; 61:367–74; Emerson E. Self-reported exposure to disability is associated with poorer self-reported health and wellbeing among adults with intellectual disabilities in England: cross sectional survey. *Public Health*. 2010; 124.

Thank you for providing these references. We now include the Emerson (2010) reference in our Introduction section (see page 5, lines 119). In our initial submission we included a 2013 reference of Alvarez-Galvez et al., that used 2010 European Social Survey data in our Introduction. The suggested 2016 reference also draws on this data. Therefore, we have not added this reference here to avoid overlap.

3. In the Method section it would be useful to quote the specific items used to identify people with self-reported disability.

In response to this suggestion we now provide further detail on how we ascertained self-reported disability on pages 7 (lines 155-162) of the manuscript.

4. Given that the association between age and disability is clearly non-linear, would not it be more appropriate to treat age as a categorical variable in the analyses? I assume that income was treated as a continuous variable and that education was treated as a categorical variable, but this should be stated.

We agree that the relationship between age and disability is likely non-linear. However, as all of our sample had a self-reported disability and we did not include a comparison group without disability, we opted to use the continuous range of scores for age in our analyses. In response to this comment, we re-ran our models including age-squared as a covariate instead of age as a continuous score and the results were very similar. Therefore, we prefer not to revise the models presented in the manuscript. The reviewer is correct that income was entered as a continuous variable and education as a categorical variable. This has now been clarified on page 9 (lines 210-11).

5. Were sample weights used in the analyses and did the analyses use the SPSS facility for complex sample design?

Although we are aware of the range of sample weights that have been designed for Understanding Society (including weights for the extra five minutes sample), we chose not to apply weights here due to concerns regarding reduced accuracy with the smaller sample size in our study ($n=871$) compared with the extra 5 minutes sample ($n > 8000$). We now specify that our analyses are unweighted in the Methods section (see page 10, line 240).

6. The authors may be interested in the following paper that explored exposure to some of the discrimination variables in Understanding Society among participants with/without self-reported disability. Emerson, E., Krnjacki, K., Llewellyn, G., Vaughan, C., Kavanagh, A. (2016). Perceptions of safety and exposure to violence in public places among working age adults with disabilities or long-term health conditions in the UK: Cross sectional study. *Public Health* 135, 91-6.

Thank you for bringing this reference to our attention. We now include this reference in the Discussion section of the manuscript (see page 18, lines 438-40).

Reviewer 2: Daniel Pérez-Garín

I think the present research is a relevant contribution to the literature about disability discrimination. As authors say, there are almost no works assessing the impact of discrimination with a longitudinal design. However, I think there are a number of questions that need to be addressed before the manuscript is accepted for publication.

Methods

Authors do not specify how the data from The United Kingdom Household Longitudinal Study were collected.

Thank you for taking the time to review this paper. In response to this comment we now provide this information on page 6 (lines 141-43) of the Method section.

Authors should explain why did they not decide to use a previously validated measure of discrimination.

As this was a secondary data analysis, we were limited by the discrimination measure used in UKHLS. We mention the weaknesses of this measure in our discussion section (from line 420), including that it does not reflect objective encounters with discrimination, as a self-report measure it is subject to recall bias and that it is not a specific tool for assessing disability discrimination. Despite these considerations, the lack of validation for this measure is unlikely to be a major limitation since it is probable that a person's interpretation of discrimination (and attribution of this experience to disability) is likely to be an important factor for wellbeing outcomes.

Covariates

On line 145, authors say that their definition of disability did not include mental health-related impairments. However, I wonder whether they specifically excluded them from the "other" category that is mentioned on lines 197 and 198, and which types of disabilities were included in it.

This is a valid point. Unfortunately, we do not have any further detail on this category, which means we cannot be entirely certain those with mental health-related impairments were not included in this category. We now mention this limitation on page 19 (lines 448-50) of the Discussion.

Since the population from which the sample was taken was not entirely comprised by people with disabilities, and many participants were members of several ethnic minorities, I think the effect of

other types of discrimination, such as perceived racial discrimination should also be taken into account. Was this controlled for?

Thank you for this comment. In our manuscript, we adjust for ethnicity rather than perceived ethnicity discrimination to take the diverse nature of our sample into account. In terms of other types of discrimination, disability discrimination was the most commonly reported form of discrimination (13.4%) in the sample, followed by ageism (4.3%), sexism (3.9%), ethnicity discrimination (3.8%), religious discrimination (2.2%) and discrimination on the basis of sexual orientation (0.5%). This suggests that disability is the most pertinent form of perceived discrimination in the sample. We have now added information on the prevalence of other types of discrimination to the manuscript (see page 11, lines 258-61).

While we acknowledge that disability discrimination may interact with other forms of discrimination to influence wellbeing, we believe that the issue of intersectionality between disability discrimination and other forms of discrimination would be best explored in a separate manuscript. We now acknowledge the point raised by the reviewer and discuss the importance of this future work on pages 18 (lines 431-34).

Results

I think the criterion used to categorize participants who perceived discrimination is unclear. There were several different questions. Was it enough to answer "yes" to any of them? Did they have to answer "yes" to at least half of them?

We are sorry this was not clearer in our initial submission. We now clarify this on page 7 (line 174).

Tables

In the file I reviewed, tables 1 and 2 overlap, as do tables 4 and 5, thus, I could not visualize them correctly and was not able to review them.

We apologise for this. This formatting issue seems to have occurred when the word document files were converted to pdf during the manuscript submission process making the tables difficult to read.

Reviewer 3: Christine Fekete

Thank you for giving me the opportunity to review the above mentioned manuscript. The study investigates cross-sectional as well as prospective associations of discrimination due to disability and a range of health-related outcomes. The manuscript is clearly structured, well-written and touches a topic of high relevance. Please find my comments below.

- General: The outcomes of the study are rather diverse, ranging from measures assessing psychological distress (GHQ), general mental health (SF-12) and life satisfaction to self-reported (general) health. Terms to describe the 'overall outcome' of the study are inconsistently used ('mental health', 'mental wellbeing', 'mental health & wellbeing outcomes', 'emotional wellbeing' in the Tables...) and although concepts correlate to some extent, it seems problematic to use 'mental health' as overall term to encompass life satisfaction & self-reported health. Could the authors think of

a rationale of why general health or life satisfaction are subsumed under mental health or adapt terminology to better capture the diversity of concepts?

Thank you for taking the time to review this paper. We agree that we could have been more consistent with the terminology throughout the manuscript. We think that wellbeing is likely the term that most accurately reflects our measures as a whole (self-reported health, as well as mental health outcomes). We have changed the terminology throughout the paper to reflect this.

Article summary

- The authors state that they examine 'changes in mental wellbeing'. However, change is not assessed with the prognostic model used. Please revise sentence accordingly.

Thank you for pointing this out. We have now removed the term "change" from the Article Summary.

Introduction

- I wonder whether the authors could link the concepts of disability to their argumentation in the second paragraph. I.e., 'these practical difficulties are mainly activity limitations and participation restrictions that might be aggravated by environmental factors, such as discrimination.'

In response to this comment we have amended the opening line of the second paragraph (page 4, line 85-86).

- line 79f. Would be worthwhile to shortly discuss reasons for the increase in disability prevalence, is population ageing the major driver? Same for line 91f., would be interesting to have an idea why disability discrimination is rising, is there a change in social climate or because disability prevalence has risen?

We agree the reasons behind the rise in disability and disability discrimination are interesting. We now include a sentence speculating that the rise in self-reported disability may be due to ageing (line 81). It is unclear why people perceive disability discrimination to be on the rise in Europe, as the Eurobarometer surveys do not survey the same participants over time. We now acknowledge this on lines 93-4.

- The meta-analytic review by Schmitt et al. seems to have included only 8 studies that included physical illness/disability discrimination. Suggestion to mention this low number as another argument for conducting this study (preventing that the vast number of 328 studies mentioned (line 104) leaves the reader with the question of why it is worthwhile to do another study on discrimination and mental health).

Thank you for this suggestion. This is an important point. We have altered the text on page 5 (lines 107-8) to make this point clear to the reader.

Methods

Study population:

- Please add some more details on the UKHLS study, i.e. what was the rationale for this study; what were its inclusion criteria; what was the sampling frame; What was the response rate of wave 1 and what was the response rate/drop-out rate at wave 5?

In response to this comment we now provide information on the study rationale (page 6, lines 138-40), the sampling frame (page 6, from line 143) and the response rate at wave 1 (see page 7, lines 161-2). We provided information on our study specific drop out rate in our initial submission (see page 7, lines 163).

- Is the 'ethnic minority boost sample' the same as the 'extra 5 minutes sample'?

The majority of the extra 5 minutes sample came from the ethnic minority boost. The difference is that the extra 5 minutes sample also included a small white comparison group. We have clarified this on page 7 (lines 152-3).

Perceived disability discrimination:

- Wonder whether the authors thought about using more fine graded information in order to better depict the frequency of discrimination (i.e. summing up the discrimination experiences in different setting). This would enable the investigation of a dose-response relationship between discrimination and mental health what would add to the value of the paper.

Thank you for this comment. This is an interesting suggestion. In our second sensitivity analysis we tested whether any particular type of discriminatory behaviour (feeling unsafe, avoiding somewhere, being insulted or physically attacked) was driving the association between perceived disability discrimination and wellbeing outcomes. We found that associations were similar when removing each of the discriminatory behaviours in turn, suggesting that the links between perceived disability discrimination and wellbeing outcomes were similar, regardless of the type of discrimination reported. For this reason, we chose in this manuscript to present perceived disability discrimination as a simple binary predictor rather than assessing it as a continuous measure. The measure of frequency of discrimination in UKHLS is limited as it does not assess the frequency of encounters in time (e.g. daily, weekly, monthly etc.). We are unsure whether the frequency of discrimination reporting across settings would be a useful metric as it depends on whether someone would visit all locations. We now mention our lack of information on frequency in the Discussion section (see page 18, lines 441-44).

Covariates:

- line 182: 'covariates that are likely relevant': Would be good if the covariates included in the models would have been empirically tested for their association with the predictors and the outcomes in that sample. Including covariates that are not associated with both, the predictors and the outcomes reduce the precision of the estimates and could be removed from the models.

We are sorry we were not clearer with our language in our initial submission. We selected our covariates a priori based on previous literature in line with guidance on defining statistical analysis plans in advance in observational research (e.g. Greenland, S., & Pearce, N. (2015). *Annual Review of Public Health*, 36, 89-108; Thomas, L., & Peterson, E. D. (2012). *JAMA*, 308(8), 773-4). As our models include fairly minimal adjustment for sociodemographic factors we prefer to retain this

covariate selection strategy. We now clarify our covariate selection strategy on page 9, line 202 and provide further literature-based justification for our covariates (page 9, lines 203- 208).

- line 197: It remains unclear what kind of impairments were classified into the category 'other' and given that around 40% of the sample belong to this category, it would be worthwhile thinking about a more fine-graded classification (if information is available). Table 1 shows for example that the majority of persons perceiving discrimination belong to the category 'other' (65%), it would thus be important to better characterize this group.

We agree this is an important point. Unfortunately, we do not have any further information on this category. We acknowledge this limitation in the discussion section (see page 18, line 441-4).

- Given that this sample consists of mostly ethnic minorities (line 141), the double burden of being part of an ethnic minority and having a disability is not accounted for in analysis. Please consider adjusting analysis for 'discrimination due to ethnicity' or critically discuss this issue.

Thank you for this comment. In our manuscript, we adjust for ethnicity rather than perceived ethnicity discrimination to take the diverse nature of our sample into account. In terms of other types of discrimination, disability discrimination was the most commonly reported form of discrimination (13.4%) in the sample, followed by ageism (4.3%), sexism (3.9%), ethnicity discrimination (3.8%), religious discrimination (2.2%) and discrimination on the basis of sexual orientation (0.5%). This suggests that disability is the most pertinent form of perceived discrimination in the sample. We have now added information on the prevalence of other types of discrimination to the manuscript (see page 11, lines 258-61). Additionally, in line with this comment we discuss the importance of investigating the interaction of disability and ethnicity discrimination on wellbeing outcomes on page 17 (lines 431-36).

Statistical analysis:

- Please specify the how missing values were treated. It seems that these are only full case analysis. It is strongly recommended to use multiple imputation to account for item-non-response.

Thank you for this comment. We chose to carry out a complete case analysis due to the extent of missing data at the follow-up wave 5 (431 cases lost to follow-up/ 871 participants at wave 1 = 49.48%). We now clarify this in the statistical analysis section on page 10, lines 236-7. In our initial submission, we addressed the issue of missing data in our first sensitivity analysis, where we show that the results of the cross-sectional analyses are similar in those who were and were not lost to follow-up. In response to this comment, we now provide an additional sensitivity analysis with imputed data at wave 5. We refer to this analysis on page 11, lines 251-53 and page 13-4 lines 320-25. We have placed this information in a supplementary table due to the journal limit of a maximum of 5 tables/figures per manuscript.

- It is a pity that the predictor variable is dichotomized. Seeing a categorical variable summing up the discrimination experiences would enable to detect a kind of a dose-response relationship between discrimination and the health outcomes.

Please see our response to your earlier comment (" Perceived disability discrimination:

- Wonder whether the authors thought about" ...).

Results

- Tab 1: Basic description of outcome variables should be placed in Table 1 rather than in Table 2 and column with the characteristics of the total population would be helpful. Adding indicators of which wave data come from for outcome variables would help the reader.

In response to this suggestion, we have added a column with the characteristics of the overall sample and have included unadjusted wellbeing data from wave 1 and 5 in this table.

- Tab 2: It is difficult to read the table as one is used to see the outcome variable in the top row instead of predictor variable; please put outcomes in the top row and the predictors in the first column; results of prospective analysis should add information on the wave at which data were collected.

Thank you for this comment. We have now removed the descriptive information from this table (in line with the comment above). In response to this comment, we have moved the outcome variables to the top row and predictors to the first column. We have also added the wave in which the prospective data were collected to the table. For consistency, we have also edited the presentation of Table 3.

- lines 263ff: Would help the reader to add information on the measurement time point, e.g. "...those who reported perceived disability discrimination at wave 1 had higher levels of" "Those who reported disability discrimination at wave 1 had slightly lower life satisfaction at wave 5..."

In response to this comment we have added information on the wave the data was collected at to the "Prospective associations between perceived disability discrimination and wellbeing" section (page 12-13).

Discussion

- May discuss the fact that persons with severe cognitive impairments were not included in the study sample as the survey inclusion criteria was probably prerequisite for study participation was the ability to respond to the questions.

Thank you for this suggestion. We have now added this limitation to page 18, lines 445-6.

- lines 313ff: Another possibility for the null finding might also be that the measurement of perceived disability discrimination is rather crude in terms of frequency and burden. One can assume that people facing discrimination repeatedly and frequently are more affected in terms of health than someone who was exposed to a single situation of discrimination over one year... Therefore, a more fine-grained analysis using kind of a sumscore over all 4 discrimination variables might reveal different findings.

We now mention this limitation in the Discussion section (see page 18, lines 441-44).

- The interpretation of results in paragraphs 3 and 4 is interesting, thank you. Are there other studies showing a more immediate or direct effect of discrimination on affectivity? Experiencing discrimination

is highly likely to increase negative affectivity, such as anger, sadness, and that repeated experiences may increase the risk of psychological distress or depression if adequate coping is not given?

This is an interesting possibility. Experimental studies that have investigated other forms of discrimination (e.g. sexism/racism) show that laboratory-based discrimination protocols are correlated with increases in negative affectivity. However, we did not come across any studies investigating the long-term impact of these exposures in our review of the literature.

- Last paragraph: it is a bit unlucky to close the paper with this paragraph as it again contains a discussion on mechanisms underlying the observed association. Suggestion to place this (interesting) paragraph before the strength/limitations section and ending the article with concrete suggestions for individual-level interventions (one of which can then be 'strengthening social support').

In response to this comment, we have removed the reference to mechanisms in this paragraph. We have not moved the paragraph as it now only makes reference to potential ways to intervene as suggested.

Reviewer 4: Anne Kavanagh

This is a well written paper which makes an important contribution to the literature. To improve the manuscript I suggest the following amendments:

1. There is considerable loss to follow-up but the authors do not discuss this in the Discussion. The sensitivity analysis of the cross-sectional data is reassuring but does not ensure that selection bias due to low retention is not a major problem longitudinally. This should be commented on in the Discussion.

Thank you for taking the time to review this manuscript. In response to this comment we now mention this limitation in the Discussion on page 17, lines 414-18. We also responded to a similar comment from another reviewer (Reviewer 3) and have provided an additional sensitivity analysis with imputed data at wave 5. We refer to this analysis on page 11, lines 251-53 and page 13-4 lines 320-25.

2. I was not clear why the second sensitivity analysis was conducted.

We are sorry the rationale for this analysis was not clearer. To assess discrimination participants were asked whether they had a) felt unsafe b) avoiding going to c) been insulted or d) had been physically attacked in seven different settings. If they answered yes to any one of these questions they were treated as a case of discrimination. We have now clarified this on page 7 (line 174). In our sensitivity analysis we wanted to see whether any of the four types of discrimination (i.e. feeling unsafe, avoiding somewhere, being insulted or being physically attacked) were driving the associations between disability discrimination and the wellbeing outcomes. We now provide some further information on this on pages 10-11 (lines 246-51) of the manuscript.

3. There was no rationale for testing of interactions. Unless these are incorporated into the research questions I suggest they are deleted.

In response to this comment we have deleted the information on the testing of interactions from the statistical analyses section.

VERSION 2 – REVIEW

REVIEWER	Eric Emerson Lancaster University, UK
REVIEW RETURNED	21-Jan-2020

GENERAL COMMENTS	Suitable for publication
--------------------------

REVIEWER	Daniel Pérez-Garín Centro de Educación Superior Cardenal Cisneros, Spain
REVIEW RETURNED	10-Feb-2020

GENERAL COMMENTS	I am overall satisfied with the changes and answers given by the authors and would advise the publication of this manuscript.
---

REVIEWER	Christine Fekete Swiss Paraplegic Research, Nottwil, Switzerland University of Lucerne, Department of Health Sciences & Medicine, Lucerne, Switzerland
REVIEW RETURNED	21-Jan-2020

GENERAL COMMENTS	Thank you very much for the careful revision of the manuscript. All concerns were addressed adequately.
---